# Evaluation of Hyperspectral Monitoring Model for Aboveground Dry Biomass of Winter Wheat by Using Multiple Factors

**Chenbo Yang, Jing Xu, Meichen Feng *** , **Juan Bai, Hui Sun, Lifang Song, Chao Wang, Wude Yang, Lujie Xiao, Meijun Zhang and Xiaoyan Song**

College of Agriculture, Shanxi Agricultural University, Jinzhong 030801, China
* Correspondence: fmc101@sxau.edu.cn

**Abstract:** The aboveground dry biomass (AGDB) of winter wheat can reflect the growth and development of winter wheat. The rapid monitoring of AGDB by using hyperspectral technology is of great significance for obtaining the growth and development status of winter wheat in real time and promoting yield increase. This study analyzed the changes of AGDB based on a winter wheat irrigation experiment. At the same time, the AGDB and canopy hyperspectral reflectance of winter wheat were obtained. The effect of spectral preprocessing algorithms such as reciprocal logarithm (Lg), multiple scattering correction (MSC), standardized normal variate (SNV), first derivative (FD), and second derivative (SD); sample division methods such as the concentration gradient method (CG), the Kennard–Stone method (KS), and the sample subset partition based on the joint X–Y distances method (SPXY); sample division ratios such as 1:1 (Ratio1), 3:2 (Ratio2), 2:1 (Ratio3), 5:2 (Ratio4), and 3:1 (Ratio5); dimension reduction algorithms such as uninformative variable elimination (UVE); and modeling algorithms such as partial least-squares regression (PLSR), stepwise multiple linear regression (SMLR), artificial neural network (ANN), and support vector machine (SVM) on the hyperspectral monitoring model of winter wheat AGDB was studied. The results showed that irrigation can improve the AGDB and canopy spectral reflectance of winter wheat. The spectral preprocessing algorithm can change the original spectral curve and improve the correlation between the original spectrum and the AGDB of winter wheat and screen out the bands of 1400 nm, 1479 nm, 1083 nm, 741 nm, 797 nm, and 486 nm, which have a high correlation with AGDB. The calibration sets and validation sets divided by different sample division methods and sample division ratios have different data-distribution characteristics. The UVE method can obviously eliminate some bands in the full-spectrum band. SVM is the best modeling algorithm. According to the universality of data, the better sample division method, sample division ratio, and modeling algorithm are SPXY, Ratio4, and SVM, respectively. Combined with the original spectrum and by using UVE to screen bands, a model with stable performance and high accuracy can be obtained. According to the particularity of data, the best model in this study is FD-CG-Ratio4-Full-SVM, for which the $R^2_c$, $RMSE_c$, $R^2_v$, $RMSE_v$, and RPD are 0.9487, 0.1663 kg·m$^{-2}$, 0.7335, 0.3600 kg·m$^{-2}$, and 1.9226, respectively, which can realize hyperspectral monitoring of winter wheat AGDB. This study can provide a reference for the rational irrigation of winter wheat in the field and provide a theoretical basis for monitoring the AGDB of winter wheat by using hyperspectral remote sensing technology.

**Keywords:** aboveground dry biomass; preprocessing; sample division method; sample division ratio; dimension reduction; modeling method

## 1. Introduction

Winter wheat is a common food crop; a large number of studies have believed that irrigation can affect the growth of crops to different degrees [1,2]. The aboveground dry biomass (AGDB) of winter wheat refers to the dry weight of all existing organic substances

above the ground of winter wheat in a unit area at a certain time [3,4]. It is believed that the AGDB not only reflects the growth and development of winter wheat aboveground but also is closely related to the final yield [5–7]. At the same time, some studies have believed that irrigation could affect the AGDB. For example, Fan et al. [8] conducted two treatments of irrigation at the jointing stage and delayed irrigation by 10 days at the jointing stage on winter wheat, and they found that delaying irrigation by 10 days at the jointing stage can improve the AGDB after flowering. Therefore, obtaining the AGDB in a timely manner has important reference value for judging the growth status of winter wheat and providing reasonable suggestions for the next field irrigation management system. However, the traditional method of obtaining AGDB of winter wheat has the disadvantages of destructive sampling and a time-consuming and laborious acquisition process [9,10]. Therefore, it is of great significance to find a method to quickly obtain the AGDB to promote the development of the winter wheat industry.

The application of remote sensing technology in agriculture provides technical support for the rapid acquisition of winter wheat AGDB [11]. Among them, hyperspectral remote sensing technology has the advantages of a large number of bands and spectral information and has become one of the technical means used by many researchers to obtain the AGDB [7,9,12]. For example, Fu et al. [7] constructed a hyperspectral monitoring model of winter wheat AGDB based on spectral indices and band depth analysis, and they achieved a high monitoring accuracy, with an $R^2$ and RMSE of 0.84 and 0.177 kg·m$^{-2}$, respectively. Therefore, it is feasible to use hyperspectral remote sensing technology to obtain the AGDB of winter wheat.

The process of constructing an AGDB model by using hyperspectral remote sensing technology includes mainly the following. (1) Properly preprocess spectral data to amplify spectral characteristics and reduce noise interference. (2) According to the amount of data, reasonably divide the data into a calibration set and a validation set, eliminate the data redundancy and over-fitting of the calibration set, and avoid the problem that the sample size of the validation set is too small to effectively validate the model. (3) Because hyperspectral data have a large amount of spectral information, which contains a substantial amount of useless information, it is necessary to use a band-screening algorithm to screen the full-spectrum bands to simplify the model and reduce the computational complexity. (4) The modeling algorithm is the final step of constructing a model; it is necessary to compare the modeling effects of different modeling algorithms and select the best modeling algorithm.

Therefore, in combination with previous research experience, according to the modeling process, modeling effecting factors can be summarized mainly into five categories: the preprocessing algorithm, the sample division method of the calibration set and the validation set, the sample division ratio of the calibration set and the validation set, whether to reduce the dimension of spectral data, and the modeling method [9,13–15]. Predecessors have conducted some research on the effecting factors of these models. Most of them are about preprocessing algorithms, whether to reduce the dimension of spectral data, and the modeling method. For example, Lee et al. [16] studied the effect of the first derivative (FD) and logarithmic transformation preprocessing on the lead, zinc, and copper content in water, and the results showed that the FD had the best prediction effect on lead, while the logarithmic transformation had a better prediction effect on zinc and copper. Jia et al. [9] studied the effect of a successive projections algorithm (SPA) and synergy interval partial least squares (SIPLS) on extracting the spectral characteristics of wheat leaf biomass. The results showed that both algorithms can reduce the complexity of the model while ensuring the accuracy of the model. Zhang et al. [17] selected four modeling methods, including the partial least squares regression (PLSR), random forest regression (RFR), extreme random tree (ERT), and K-nearest neighbor (KNN) algorithms, to construct hyperspectral monitoring models of winter wheat leaf water content. The final results showed that the ERT had a good modeling effect.

In addition, many studies have also analyzed these factors at the same time. For example, Zhang et al. [17] previously used the correlation coefficient method and the

x-loading weight method to screen characteristic spectra by using PLSR, RFR, ERT, and KNN to construct the model of winter wheat leaf water content. Xu et al. [18] simultaneously studied the effects of preprocessing algorithms such as the Svitzky–Golay filter, wavelet packet transform, multiple scattering correction (MSC), and fractional order derivative and dimensionality reduction methods such as principal component analysis (PCA), multidimensional scaling, and locally linear embedding on the accuracy of hyperspectral monitoring models of soil organic matter content. Li et al. [19] analyzed the effect of two dimensionality reduction algorithms of uninformative variable elimination (UVE) and SPA, and two modeling methods of PLSR and extreme learning machine on the accuracy of the monitoring model of soil total nitrogen content.

However, the effect of the sample division method and sample division ratio of the calibration set and the validation set on the model accuracy is often ignored. However, some studies have analyzed this. Yang et al. [20] set five sample division ratios of 1:1, 3:2, 2:1, 5:2, and 3:1 for 225 soil samples according to the concentration gradient method (CG) when constructing the hyperspectral monitoring model of soil organic carbon content. The results showed that the highest model accuracy was achieved when the division ratio was 3:2. Xu et al. [21] divided 301 samples according to stratified random sampling (STRAT), the Kennard–Stone method (KS), and sample subset partition based on the joint X–Y distances method (SPXY) when using hyperspectral technology to classify mangrove species. The results showed that the classification effect of STRAT was better. To sum up, above five category factors all affect the accuracy of the hyperspectral monitoring model to varying degrees.

When the predecessors used hyperspectral technology to quantitatively monitor the AGDB of winter wheat, they generally analyzed the effect of one or more of these factors on the accuracy of the model. Bao et al. [12] reduced the complexity of the model by constructing spectral indexes and extracting spectral characteristic parameters, and they achieved good results. The maximum RMSE of the model was 66.403 g·m$^{-2}$. Li et al. [22] constructed a three-band spectral index when monitoring the AGDB, and they finally screened out three bands of 560 nm, 738 nm, and 806 nm that had a strong relationship with the AGDB. Fu et al. [23] constructed a monitoring model of AGDB based on two spectral preprocessing algorithms of continuum removal (CR) and FD, and three modeling methods of principal component regression, PLSR, and stepwise multiple linear regression (SMLR). The results showed that the model constructed by CR combined with PLSR had the highest accuracy.

It can be seen from the analysis of previous studies that, when constructing the hyperspectral monitoring model of winter wheat AGDB, the predecessors rarely simultaneously considered the effect of above five category factors on the accuracy of the winter wheat AGDB model. Based on this, this study sets five kinds of preprocessing for the original spectrum, including reciprocal logarithm (Lg), MSC, standardized normal variate (SNV), FD, and second derivative (SD). Then, CG, KS, and SPXY methods are used to divide all samples into a calibration set and a validation set according to the division ratios of 1:1 (Ratio1), 3:2 (Ratio2), 2:1 (Ratio3), 5:2 (Ratio4), and 3:1 (Ratio5). Finally, based on the full-spectrum band and the band screened by UVE, respectively, the hyperspectral monitoring models of winter wheat AGDB are constructed by using PLSR, SMLR, artificial neural network (ANN), and support vector machine (SVM).

The purpose of this work is to (1) analyze the effect of irrigation on AGDB of winter wheat; (2) analyze the correlation between the original spectrum and preprocessing spectra and the AGDB of winter wheat and determine the noise-reduction effect of different preprocessing algorithms; (3) analyze the division effect of three sample division methods and five sample division ratios; (4) analyze the effect of UVE algorithm in screening bands; and (5) analyze the modeling effect of four modeling algorithms. Finally, through comprehensive comparison, this work aims to select the best spectral preprocessing algorithm, sample division algorithm, sample division ratio, and modeling method for hyperspectral monitoring models of winter wheat AGDB and determine whether dimension reduction is

required in the modeling process. This study can provide a theoretical basis for the rational irrigation of winter wheat and the rapid and non-destructive acquisition of the AGDB of winter wheat.

## 2. Materials and Methods

### 2.1. Experimental Design

The experiment was conducted in the agricultural station of Shanxi Agricultural University from October 2020 to July 2022. The experimental procedure involved building a water pond according to FAO standards. Each plot was a 2 m × 3 m rectangle with 15 plots in total. The winter wheat variety "JinTai 182" was planted with a row spacing of 20 cm. A total of 5 water treatments was set: T1 (No irrigation), T2 (Jointing stage), T3 (Jointing stage + Flowering stage), T4 (Jointing stage + Later grain-filling stage), and T5 (Jointing stage + Flowering stage + Later grain-filling stage). The irrigation amount was 60 mm each time. Each treatment was repeated 3 times by using a completely random design. The overwintering water and the reviving water were uniformly irrigated. Nitrogen, phosphorus, and potassium fertilizers were used as the base fertilizer and were applied once before sowing. The fertilization standard was 150 kg·hm$^{-2}$ for nitrogen (N), 120 kg·hm$^{-2}$ for phosphorus ($P_2O_5$), and 120 kg·hm$^{-2}$ for potassium ($K_2O$). The nitrogen source was urea, the phosphorus source was calcium superphosphate, and the potassium source was potassium sulfate. Other field management was consistent with that of local farmers.

### 2.2. Data Acquisition

Canopy hyperspectral data were measured with a Field-Spec 3.0 spectrometer (Produced by the ASD Company of the United States). The acquisition band range was 350–2500 nm. The spectral sampling interval between 350 and 1000 nm was 1.4 nm, and the spectral resolution was 3 nm. The spectral sampling interval between 1000 and 2500 nm was 2 nm, and the spectral resolution was 10 nm. The field angle was 25°. Spectral acquisition was carried out in sunny, cloudless, or windless weather or in wind less than Grade 3. For a certain plot, a position with uniform growth was selected to ensure that the collected spectrum was representative. The probe was placed at 1 m above the canopy during measurement, and 10 spectral curves were measured. Finally, the average of the 10 spectral curves was calculated as the final spectrum of this plot. Before each measurement, the whiteboard was used for calibrating.

The AGDB was measured by the drying–weighing method. Specifically, winter wheat plants with a row length of 10 cm were collected, the roots were removed, and the plants were placed into an oven. The samples were baked at 105 °C for 30 min, then the temperature was adjusted to 80 °C and dried to a constant weight. Finally, the weight was converted to the weight per unit area, and the unit was kg·m$^{-2}$.

### 2.3. Data Analysis Method

Lg can reduce the influence of multiplicative factors caused by light transformation [24,25]. MSC can eliminate the scattering effect caused by uneven sample distribution [26]. SNV can eliminate the influence of diffuse reflection spectrum [27]. FD and SD can reduce noise interference and improve the sensitivity of the spectral information [28,29].

The CG method is a method used to classify samples according to dependent variables [3,13]. In this study, the specific steps were to sort the AGDB data from small to large and to divide the samples into a calibration set and a validation set in certain ratios. For example, a ratio of 3:2 means that among 5 consecutive samples, 3 samples were selected into the calibration set, and other 2 samples were selected into the validation set. The KS method is a method to classify samples according to independent variables. In this study, the Euclidean distance between each spectrum and the rest of each spectrum was calculated. First, the two samples with the largest Euclidean distance were selected as the calibration set, and then the Euclidean distance was calculated between the remaining samples and the calibration set samples. The samples were selected with the smallest and largest distances

to enter the calibration set, and so on, until enough samples were obtained to enter the calibration set [14,30]. The SPXY method was proposed by Galvão et al. [31] based on the KS method. Its sample selection process is similar to the KS method, but in the process of calculating Euclidean distance, both independent and dependent variables are considered [14,32,33]. In this study, the ratios of 5 calibration sets and validation sets were set, which were 1:1 (Ratio1), 3:2 (Ratio2), 2:1 (Ratio3), 5:2 (Ratio4), and 3:1 (Ratio5), respectively.

UVE is a variable selection algorithm based on the PLSR, which screens bands according to the stability of the PLSR model regression coefficients [34,35].

PLSR is a modeling method integrating PCA, canonical correlation analysis, and multiple linear regression analysis [13,34]. SMLR is an algorithm that can extract significant variables to construct a linear regression model [36,37]. ANN is an algorithm for mining data relationships developed from simulating the neural system of animals [37,38]. SVM was originally used to solve classification problems, and now it is widely used to mine data relationships [39,40].

In this study, Microsoft Excel 2021 was used for data sorting. SPSS Statistics 26 was used for difference analysis. Unscrambler 9.7 was used for the preprocessing of original spectral data. The CG method, KS method, SPXY method, UVE, PLSR, SMLR, ANN, and SVM were implemented in MATLAB 2021a. Origin 2021 was used for drawing. The determination coefficient ($R^2$), root mean square error (RMSE), and relative analysis error (RPD) were used to evaluate the accuracy of the model.

## 3. Results

### 3.1. Variation in AGDB and Spectral Reflectance under Different Treatments

Figure 1 shows the change rule of the AGDB of winter wheat in the two-year experiment. It can be seen from the figure that with the growth of winter wheat, there was a certain difference in the change trend in AGDB in the two-year experiment. Among them, the experiment in 2021 mainly showed a rising trend first and reached the highest at the early grain-filling stage and then fell, while the experiment in 2022 showed a trend of gradually rising to the flowering stage and reached basic stability. In the two-year experiment, the AGDB of the regreening stage had certain differences under different treatments, but the differences were not significant. After the first irrigation treatment, the AGDB of the T2, T3, T4, and T5 treatments at the jointing stage had no significant difference but were significantly higher than that of the T1 treatment. At the booting stage and heading stage, the AGDB of the T2, T3, T4, and T5 treatments after irrigation were higher than that of the T1 treatment, but the difference reached a significant level only at the booting stage in 2021. At the flowering stage, the AGDB increased further. Among them, the T3 and T5 treatments had the largest increase after the second irrigation treatment. There was no significant difference between the T2 and T4 treatments in the two-year experiment, but it was significantly higher than the T1 treatment. At the early grain-filling stage, the AGDB of the T3 and T5 treatments in the 2022 experiment were still the highest, but the differences between the treatments were not significant in the 2021 experiment, and the change characteristics were disordered. After the third irrigation, the AGDB of T4 and T5 treatments were higher than that of the other treatments, but there was no significant difference with the T2 and T3 treatments; it was only that they were significantly higher than the T1 treatment. In the maturation stage, the AGDB changes with different treatments were also different. In 2021, it reached the highest under the T3 treatment, and in 2022, it reached the highest under the T5 treatment. The AGDB of the T2, T3, T4, and T5 treatments were higher than that of the T1 treatment, but there was no significant difference among the treatments.

Figure 2 shows the spectral reflectance curve under different irrigation treatments. The spectral bands in the range of 350–399 nm, 1351–1399 nm, 1801–1950 nm, and 2451–2500 nm were eliminated in order to reduce the effect of factors such as moisture in the air. It can be seen from the figure that the original spectrum of winter wheat canopy basically increased first and then decreased with the increase in wavelength. Among them, a small reflection

peak was formed near 550 nm in the visible light area, a near-infrared reflection platform was formed within 780–1100 nm, and two obvious absorption valleys were formed near 1000 nm and 1450 nm, which conforms to the basic characteristics of spectral reflectance of the green plant canopy. However, at the same wavelength, there were certain differences in spectral reflectance between different treatments. Taking the near-infrared reflection platform as an example, we can see that the spectral reflectance of the T1 treatment without irrigation was the lowest. The T4 and T2 treatments irrigated once at the jointing stage were higher than the T1 treatments, and the T3 and T5 treatments irrigated twice at the jointing stage and flowering stage were the highest.

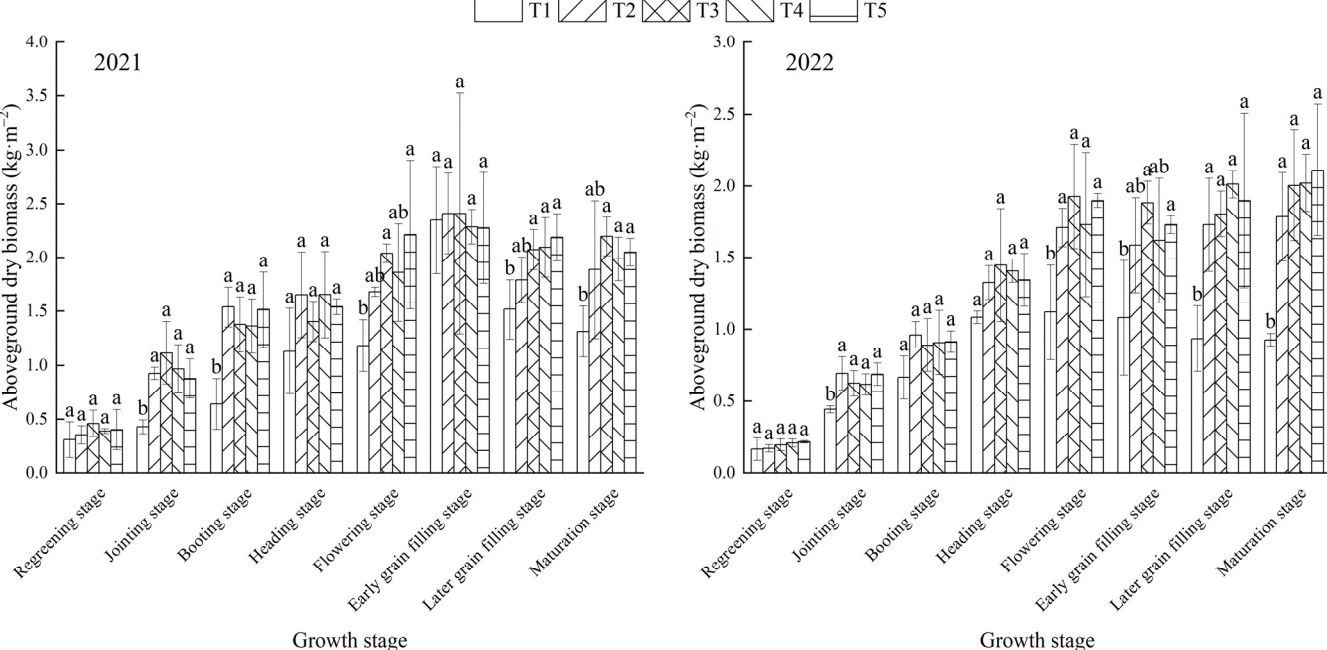

**Figure 1.** Changes of AGDB of winter wheat in two-year experiment. Note: T1: No irrigation. T2: Irrigation at jointing stage. T3: Irrigation at jointing stage and flowering stage. T4: Irrigation at jointing stage and later grain-filling stage. T5: Irrigation at jointing stage, flowering stage, and later grain-filling stage. The error bars are the standard deviation. The Tukey method was used for multiple comparison. Lowercase letters indicate that the difference reaches a significance level ($p < 0.05$), that is, in five treatments at a certain growth stage, having the same letter between any two treatments means that the difference between the two treatments is not significant, while not having the same letter means that the difference between the two treatments is significant.

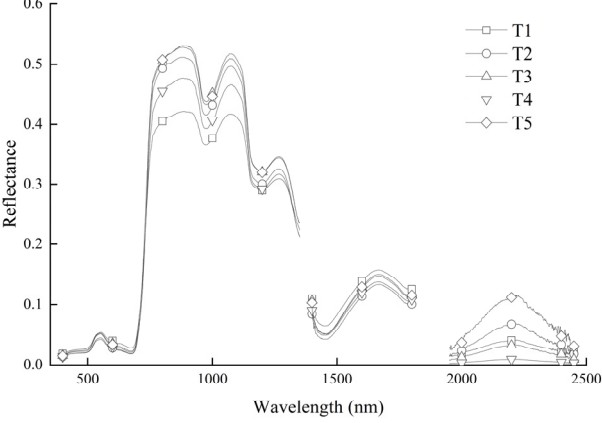

**Figure 2.** Change rule of winter wheat canopy spectrum under different treatments. Note: Take the flowering stage in the 2021 experiment as an example.

## 3.2. Effect of Preprocessing Algorithm on Spectral Reflectance

In this study, Lg, MSC, SNV, FD, and SD were preprocessed for the original spectral (R) curve, and the results are shown in Figure 3. (Take T3 treatment at flowering stage in 2021 as an example; its AGDB is 1.3405 kg·m$^{-2}$, which is closest to the average value of AGDB of all samples in this study.). From the preprocessing spectrum, it can be seen that the changing trend in the Lg transformation spectrum was basically opposite to that of R. The changing trends of MSC and SNV transformation spectra were basically the same as that of R. The FD and SD transformation spectra basically lost the basic characteristics of R, but from the geometric significance of FD and SD, we can see that FD and SD transformation can show the changes in R that are difficult to recognize by the naked eye.

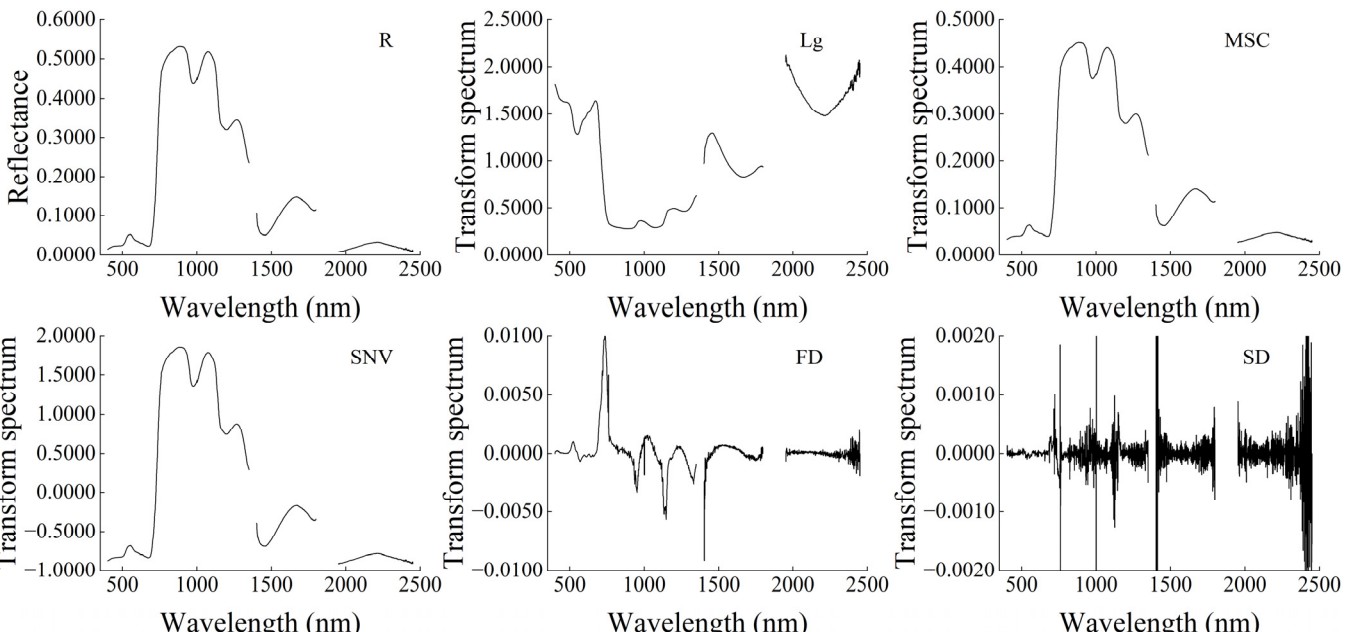

**Figure 3.** Original spectrum and preprocessing spectra. Note: R, Lg, MSC, SNV, FD, and SD are original spectrum, reciprocal logarithm transformation spectrum, multiple scattering correction transformation spectrum, standardized normal variate transformation spectrum, first derivative transformation spectrum, and second derivative transformation spectrum, respectively. The same below.

Figure 4 shows the correlation between R and preprocessing spectra and AGDB of all samples. The correlation coefficient between R and AGDB ranged from −0.2332 to 0.2640, and the preprocessing algorithms can improve the correlation between spectral reflectance and AGDB to varying degrees. There were differences in the band positions when each spectrum had a significant correlation with AGDB. In general, FD preprocessing had the best effect on improving the correlation between original spectral reflectance and AGDB, followed by MSC, SD, SNV, and Lg. In addition, there were some differences in the band positions when the correlation reached the highest, which were 1400 nm, 1479 nm, 1083 nm, 741 nm, 797 nm, and 486 nm, respectively. It can be seen that each preprocessing algorithm has different effects in reducing noise interference and amplifying useful information in the spectral curve.

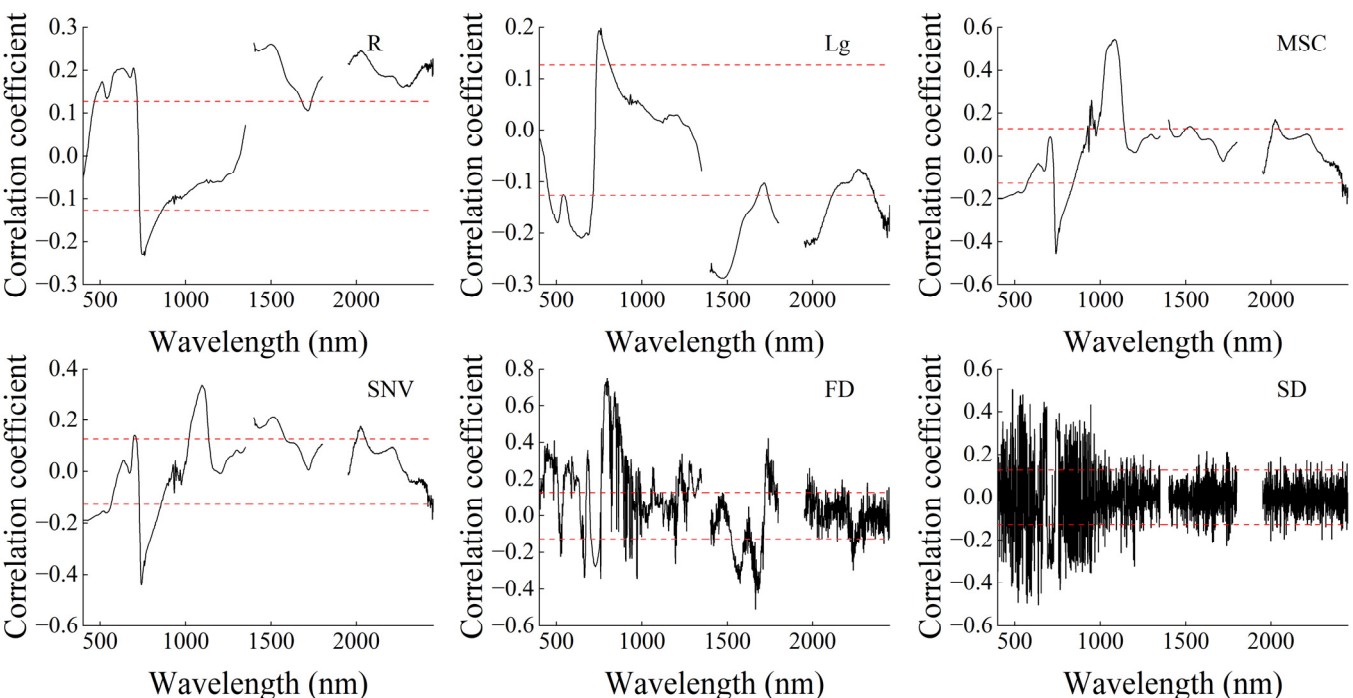

**Figure 4.** Correlation between R and preprocessing spectra and AGDB. Note: The red line is the critical value of significance (*p* < 0.05).

### 3.3. Descriptive Statistics

In this study, 3 sample division methods, CG, KS, and SPXY, were used to divide 240 samples into the calibration set and the validation set according to the ratio of 1:1, 3:2, 2:1, 5:2, and 3:1. At the same time, for the convenience of subsequent analysis, the division was based on the original spectrum. The data characteristics of the divided data set are shown in Table 1. It can be seen from the table that the maximum and minimum values of the total data set were 3.7185 and 0.0785 kg·m$^{-2}$, respectively. The data sets that were divided based on the CG and SPXY methods assigned the maximum and minimum values to the calibration set; it ensured that the range of the model was not exceeded when the model was validated, while the data sets that were divided based on the KS method assigned the maximum values to the validation set and the minimum values to the calibration set. The average value and standard deviation of the total data set were 1.3392 and 0.7196 kg·m$^{-2}$, respectively. The average value and standard deviation of all calibration sets and validation sets are close to it. The kurtosis of the total data set, the calibration sets, and the validation sets that were divided by the CG and SPXY methods, and the calibration sets that were divided by the KS method were all negative, indicating that the distribution of these data sets is slower than that of the normal distribution, while the validation sets that were divided by the KS method are steeper than that of the normal distribution. Except for the calibration sets that were divided by the KS method under Ratio1 and the validation sets that were divided by the SPXY method under Ratio1, Ratio2, and Ratio3, the skewness of other data sets was positive, indicating that these data sets had different degrees of right skewness compared with the normal distribution. Comprehensive analysis showed that different sample division methods and division ratios have different division effects.

**Table 1.** Division effect of different sample division methods and ratios.

| Division Method | Data Sets | Num | Max (kg·m⁻²) | Min (kg·m⁻²) | Avg (kg·m⁻²) | SD (kg·m⁻²) | Kurtosis | Skewness |
|---|---|---|---|---|---|---|---|---|
| | Total | 240 | 3.7185 | 0.0785 | 1.3392 | 0.7196 | −0.3388 | 0.2536 |
| | Cal-Ratio1 | 120 | 3.7185 | 0.0785 | 1.3342 | 0.7228 | −0.2125 | 0.2767 |
| | Val-Ratio1 | 120 | 3.5210 | 0.0865 | 1.3442 | 0.7193 | −0.4290 | 0.2336 |
| | Cal-Ratio2 | 144 | 3.7185 | 0.0785 | 1.3390 | 0.7199 | −0.3090 | 0.2566 |
| | Val-Ratio2 | 96 | 3.5210 | 0.0865 | 1.3395 | 0.7229 | −0.3373 | 0.2531 |
| CG | Cal-Ratio3 | 160 | 3.7185 | 0.0785 | 1.3385 | 0.7182 | −0.3585 | 0.2398 |
| | Val-Ratio3 | 80 | 3.5210 | 0.0865 | 1.3405 | 0.7268 | −0.2443 | 0.2855 |
| | Cal-Ratio4 | 172 | 3.7185 | 0.0785 | 1.3473 | 0.7319 | −0.1795 | 0.3167 |
| | Val-Ratio4 | 68 | 2.7240 | 0.1210 | 1.3188 | 0.6922 | −0.8785 | 0.0536 |
| | Cal-Ratio5 | 180 | 3.7185 | 0.0785 | 1.3355 | 0.7164 | −0.4086 | 0.2225 |
| | Val-Ratio5 | 60 | 3.5210 | 0.1210 | 1.3501 | 0.7349 | −0.0706 | 0.3477 |
| | Cal-Ratio1 | 120 | 2.8690 | 0.0785 | 1.2936 | 0.7276 | −0.9945 | −0.0191 |
| | Val-Ratio1 | 120 | 3.7185 | 0.1470 | 1.3848 | 0.7116 | 0.2538 | 0.5605 |
| | Cal-Ratio2 | 144 | 2.8690 | 0.0785 | 1.3135 | 0.7234 | −0.8989 | 0.0515 |
| | Val-Ratio2 | 96 | 3.7185 | 0.1470 | 1.3776 | 0.7158 | 0.4846 | 0.5797 |
| KS | Cal-Ratio3 | 160 | 2.8690 | 0.0785 | 1.3202 | 0.7180 | −0.8951 | 0.0396 |
| | Val-Ratio3 | 80 | 3.7185 | 0.1970 | 1.3772 | 0.7258 | 0.6863 | 0.6796 |
| | Cal-Ratio4 | 172 | 3.5210 | 0.0785 | 1.3612 | 0.7315 | −0.6462 | 0.0746 |
| | Val-Ratio4 | 68 | 3.7185 | 0.1970 | 1.2836 | 0.6907 | 0.9729 | 0.7826 |
| | Cal-Ratio5 | 180 | 3.5210 | 0.0785 | 1.3581 | 0.7348 | −0.6776 | 0.0915 |
| | Val-Ratio5 | 60 | 3.7185 | 0.1995 | 1.2823 | 0.6746 | 1.4735 | 0.8602 |
| | Cal-Ratio1 | 120 | 3.7185 | 0.0785 | 1.4037 | 0.8062 | −0.4957 | 0.2753 |
| | Val-Ratio1 | 120 | 2.6310 | 0.0865 | 1.2747 | 0.6177 | −0.7892 | −0.0241 |
| | Cal-Ratio2 | 144 | 3.7185 | 0.0785 | 1.4045 | 0.7924 | −0.5280 | 0.2215 |
| | Val-Ratio2 | 96 | 2.4780 | 0.0865 | 1.2412 | 0.5842 | −0.8944 | −0.0622 |
| SPXY | Cal-Ratio3 | 160 | 3.7185 | 0.0785 | 1.3955 | 0.7693 | −0.4498 | 0.2321 |
| | Val-Ratio3 | 80 | 2.4780 | 0.0865 | 1.2265 | 0.5967 | −0.8646 | −0.0312 |
| | Cal-Ratio4 | 172 | 3.7185 | 0.0785 | 1.3786 | 0.7626 | −0.4159 | 0.2226 |
| | Val-Ratio4 | 68 | 2.4780 | 0.1500 | 1.2396 | 0.5905 | −0.8782 | 0.0748 |
| | Cal-Ratio5 | 180 | 3.7185 | 0.0785 | 1.3834 | 0.7503 | −0.3605 | 0.2138 |
| | Val-Ratio5 | 60 | 2.4780 | 0.1500 | 1.2065 | 0.6047 | −0.9547 | 0.1391 |

Note: Num, Max, Min, Avg, SD, Cal, and Val refer to number of sample, maximum, minimum, average, standard deviation, calibration set, and validation set, respectively.

### 3.4. Screening Band Effect by Dimension Reduction Algorithm

In this study, the UVE algorithm was selected to screen hyperspectral bands, and the screening results are shown in Figure 5. It can be seen from the figure that the effect of using UVE to screen bands was different based on different preprocessing spectra. Among them, the bands that were screened based on R were distributed mainly in the range of 500–540 nm, 750–1000 nm, and 1200–1250 nm, but the bands that were screened based on the data sets that were divided via CG and SPXY methods were still distributed in the range of 1300–2450 nm, while the bands that were screened via the KS method were less in the range of 1300–2450 nm. The bands that were screened based on the Lg transformation spectrum were similar to the R results, which were distributed mainly in the range of 400–500 nm, 700–1110 nm, 1470–1530 nm, 1650–1800 nm, and 2000–2330 nm. The bands screened based on the MSC transformation spectrum were distributed mainly in the range of 500–530 nm, 750–1000 nm, 1100–1220 nm, and 1310–1350 nm. Among them, the bands that were screened based on the data sets that were divided by the KS and SPXY methods were still distributed to a small extent in the range of 1450–2450 nm. The bands that were screened based on the SNV transformation spectrum were distributed mainly in the range of 500–540 nm and 750–1250 nm, and the data sets that were divided by some sample division methods and ratios were also distributed in the range of 1400–2450 nm. The screening results of the bands that were screened based on the FD transformation spectrum were similar under each sample division method and ratio and were distributed mainly in

the range of 400–1750 nm. The bands that were screened based on the SD transformation spectrum were distributed mainly in the range of 400–1000 nm. The bands that were screened based on the data sets that were divided by the SPXY method did not include the bands in the range of 1660–2449 nm. As far as the number of screening bands is concerned, the distribution ranged from 10 to 1001. The number of screening bands that were based on the R, Lg, MSC, SNV, FD, and SD spectra was 148–344, 510–948, 76–385, 113–1001, 214–510, and 10–319, respectively. The number difference based on R is small, while the number difference based on the SNV transformation spectrum is large. It can be seen that there were certain differences in the effect of using UVE to screen bands based on different preprocessing spectra, sample division methods, and division ratios.

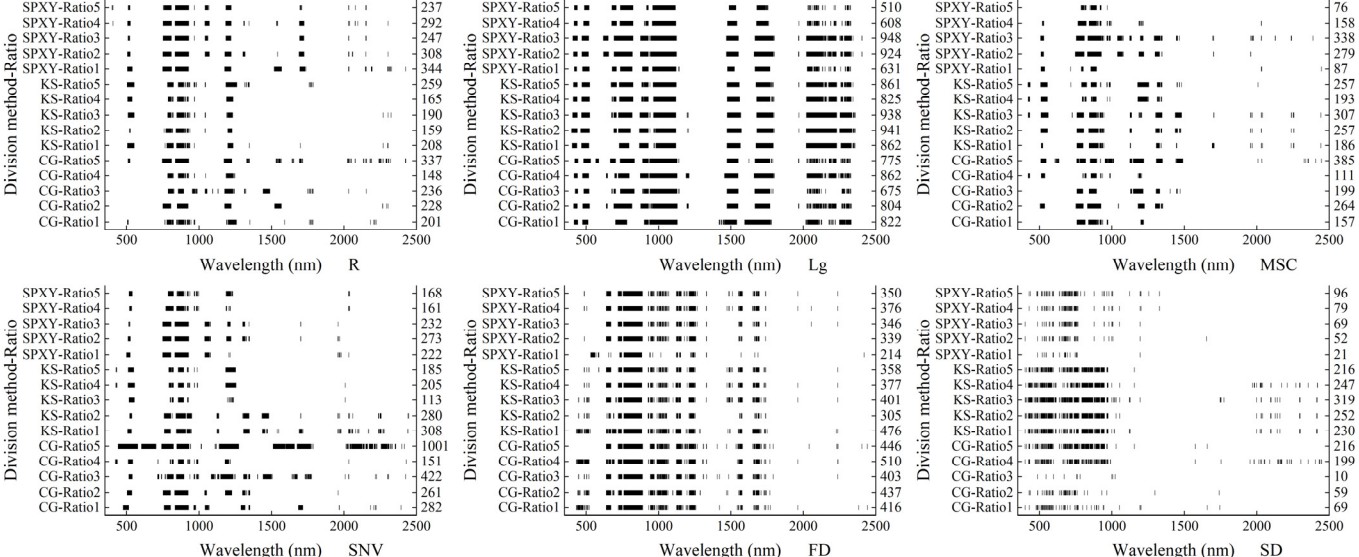

**Figure 5.** Band-screening results of UVE under different preprocessing algorithms. Note: The number on the right is the number of bands; the same is true below.

SMLR is an algorithm that can screen bands during modeling. Therefore, this paper also analyzed the bands that were used in constructing SMLR models. The SMLR modeling bands that were based on full-spectrum bands are shown in Figure 6. Unlike the UVE bands, SMLR modeling bands were mostly discrete bands. Among them, the SMLR modeling bands that were based on R were distributed mainly in the near-infrared long wave (1100–2450 nm) region, followed by the near-infrared short wave (780–1100 nm) region, and these bands were the least distributed in the visible light (400–780 nm) region. The SMLR modeling bands that were based on the Lg transformation spectrum varied greatly due to the sample division method and ratio, mainly including the bands of approximately 400 nm and 1070 nm, and the bands within the range of 1400–1800 nm. In addition, the bands that were based on the CG method did not include bands within the range of 1100–1350 nm and 1951–2450 nm. The bands that were based on the MSC transformation spectrum included mainly bands in the range of 900–1150 nm and 2400–2450 nm. In addition, bands that were based on the CG method were also generally distributed in the range of 1400–1800 nm, while bands that were based on the KS method were also distributed in the range of 550–760 nm. The bands that were based on the SNV transformation spectrum were distributed mainly in the range of 500–1500 nm and 2400–2450 nm, and they were distributed less in the range of 1500–2400 nm. The bands that were based on the FD transformation spectrum basically had uniform distribution in the full spectrum range, but the distribution in the near-infrared band (780–2450 nm) is more than that in the visible band (400–780 nm). Similar to FD, the bands that were based on the SD transformation spectrum were uniformly distributed in the full spectrum range. In terms of the number of bands, the number of SMLR modeling bands ranged from 2 to 173. Among them, the

number of bands screened based on FD and SD transformation spectra was the largest. At the same time, under the same sample division method and ratio, the number of bands that were screened based on FD and SD transformation spectra is basically the same. The number of bands that were screened based on R, Lg, MSC, and SNV transformation spectra was generally less than 50. It can be seen that there are also some differences in the SMLR modeling bands that were based on the different preprocessing spectra, sample division methods, and ratios.

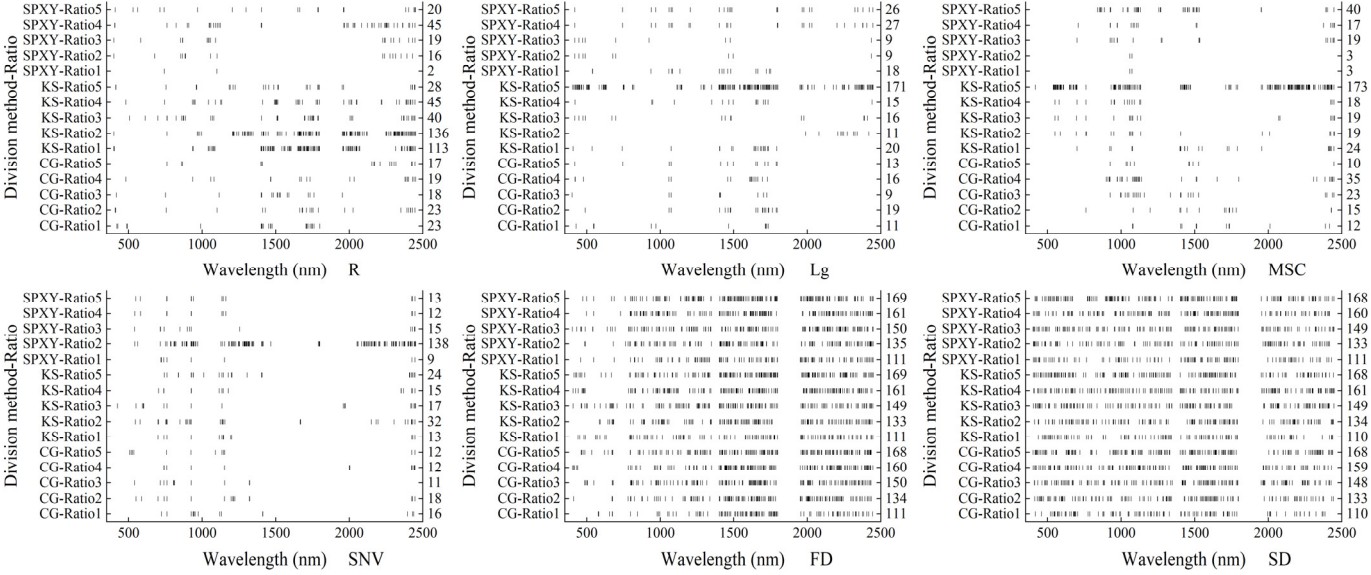

**Figure 6.** SMLR modeling bands based on full-spectrum bands.

Figure 7 shows SMLR modeling bands that were based on UVE bands (UVE-SMLR). As can be seen from the figure, the number of UVE-SMLR modeling bands is generally small, ranging from 1 to 29, in contrast with the larger number of UVE bands and SMLR modeling bands that were based on the full-spectrum bands. Among them, the number of UVE-SMLR modeling bands that were based on R ranged from 1 to 14, mainly including bands in the range of 750–1000 nm. The number of bands that were based on the Lg transformation spectrum ranged from 9 to 21, mainly including bands in the range of 420 nm, 700–1100 nm, 1480–1500 nm, and 1670–1730 nm. The number of bands that were based on the MSC and SNV transformation spectra ranged from 2 to 16 and 4 to 15, mainly including bands in the range of 760–1350 nm, and less in the range of 1400–2450 nm. The number of bands that were based on the FD transformation spectrum ranged from 4 to 27, mainly including bands of approximately 650 nm, 790–1250 nm, 1670 nm, and 2237 nm. The number of bands that were based on the SD transformation spectrum ranged from 3 to 29, mainly including the bands in the range of 470–1000 nm, while the distribution in the range of 1000–2450 nm was less.

If the SMLR algorithm is regarded as a band-screening algorithm, combined with the effect of UVE bands screening, it can be seen that there were different effects when band screening was based on different preprocessing algorithms, sample division methods, and ratios.

### 3.5. Modeling Results of Different Modeling Algorithms

After different preprocessing of the original spectrum and sample division according to different sample division methods and ratios, the hyperspectral monitoring models for the AGDB of winter wheat were constructed by using PLSR, SMLR, ANN, and SVM based on the full-spectrum band and the band screened by UVE, respectively. The accuracy of the calibration set model is shown in Figure 8. It can be seen from the figure that the $R^2_c$ values of most models are above 0.6, and the $RMSE_c$ values are below 0.5 kg·m$^{-2}$. However,

different factors had different effects on the accuracy of the calibration set model. As far as the preprocessing algorithm is concerned, the accuracy of the calibration set model based on FD and SD preprocessing was generally higher than that of other preprocessing, and the effect of FD preprocessing was better than that of SD. As far as the sample division method is concerned, the calibration accuracy of the model that was constructed based on the data set divided by the KS method was generally higher than that of the CG method and the SPXY method. As far as the sample division ratio is concerned, under the same spectral preprocessing algorithm, the model that was based on Ratio1 had a higher calibration accuracy. In terms of whether to use the UVE algorithm to screen bands, the accuracy of the models that were constructed based on the full-spectrum band was generally higher than that of the models that were constructed based on the UVE band. As far as the modeling method is concerned, SMLR and SVM were the most effective, followed by PLSR and ANN. It is worth noting that the $R^2_c$ of all SMLR models that were based on the full-spectrum band of FD and SD preprocessing, as well as R-KS-Ratio1-Full-SMLR, R-KS-Ratio2-Full-SMLR, Lg-KS-Ratio5-Full-SMLR, MSC-KS-Ratio5-Full-SMLR, and SNV-SPXY-Ratio2-Full-SMLR, reached 1, and the $RMSE_c$ was 0 kg·m$^{-2}$. According to the number of SMLR modeling bands that were based on the full spectrum in Figure 6, this may be related to the algorithmic nature of SMLR. In addition, the model accuracy of SD-SPXY-Ratio2-Full-SVM was the best, the $R^2_c$ reached 0.9997, and the $RMSE_c$ was only 0.0142 kg·m$^{-2}$.

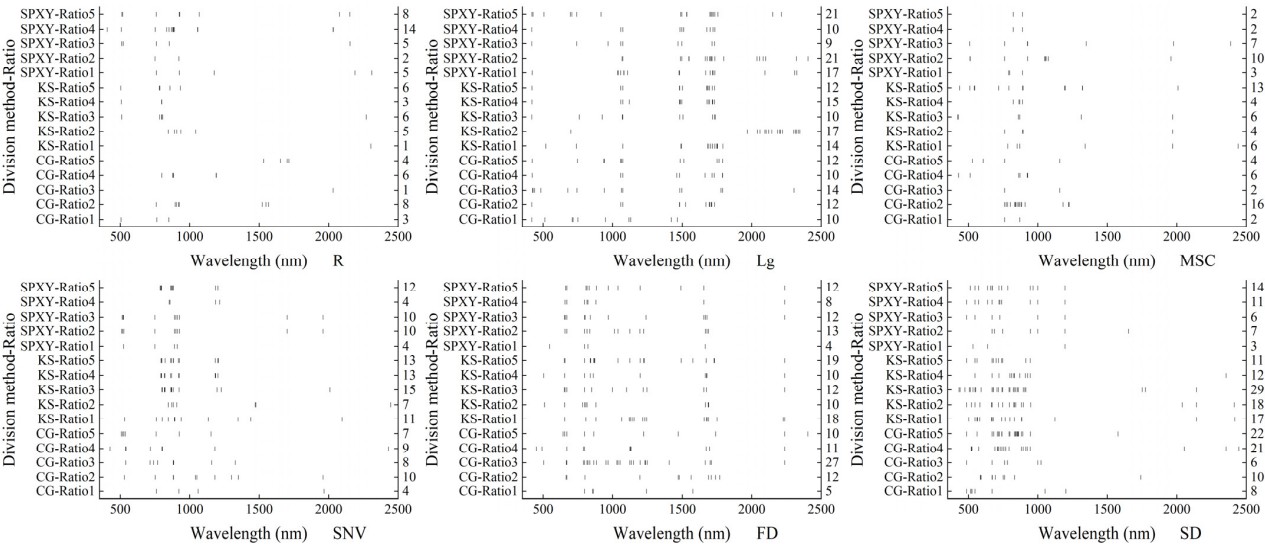

**Figure 7.** SMLR modeling bands based on UVE bands.

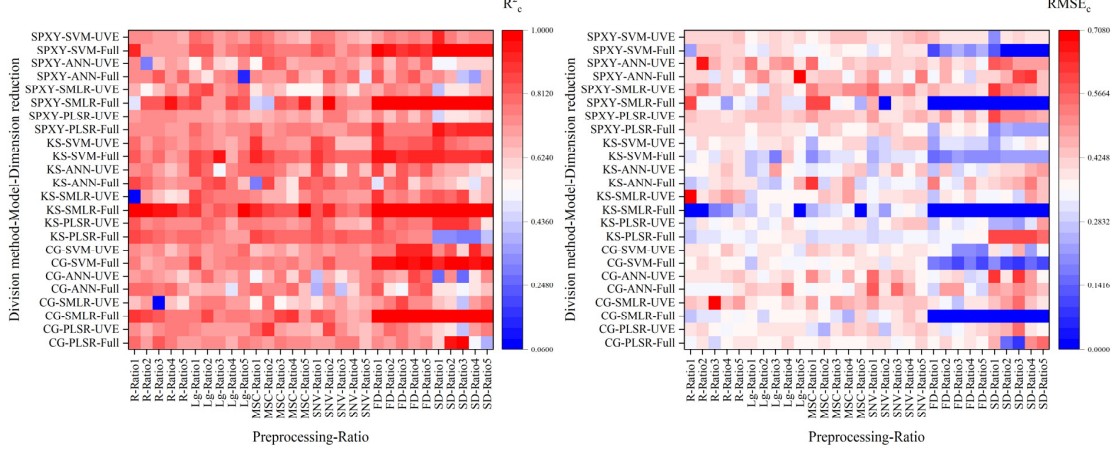

**Figure 8.** Accuracy of calibration set model.

Figure 9 shows the accuracy of the validation set model. It can be seen from the figure that the validation accuracy and calibration accuracy are quite different. Distinct from the high accuracy of the calibration model under SD preprocessing, the validation accuracy corresponding to the model was generally low, and its $R^2_v$ and RPD were generally lower than 0.6 and 1.4, respectively. In addition, the validation accuracy of models that were based on R was generally higher than that of models that were based on other preprocessing spectra. The validation accuracy of models that were constructed based on different sample division methods was different due to the modeling methods and whether dimension reduction was selected, and there was no clear rule. However, the accuracy of the models that were constructed based on the SPXY method was generally higher than those that were based on the CG method and KS method. The validation accuracy of models that were constructed based on different sample division ratios is usually higher in Ratio4 or Ratio5. Under the same conditions, the validation accuracy of models that were based on the UVE band is generally higher than those based on the full-spectrum band. The effect of modeling methods on model validation accuracy varied with different sample division methods. The PLSR and SVM models that were based on CG and SPXY are generally more accurate than SMLR and ANN models, but the models that were based on KS had higher validation accuracy only when using SVM modeling. Among all models, the FD-CG-Ratio4-Full-SVM model reached the highest validation accuracy, with $R^2_v$, $RMSE_v$, and RPD values of 0.7335, 0.3600 kg·m$^{-2}$, and 1.9226, respectively. The $R^2_c$ and $RMSE_c$ of the model were 0.9487 and 0.1663 kg·m$^{-2}$, respectively, indicating that the model had high calibration accuracy at this time. Therefore, it can ensure that the model had high prediction accuracy and stability. With the help of this model, the hyperspectral monitoring of AGDB of winter wheat can be realized.

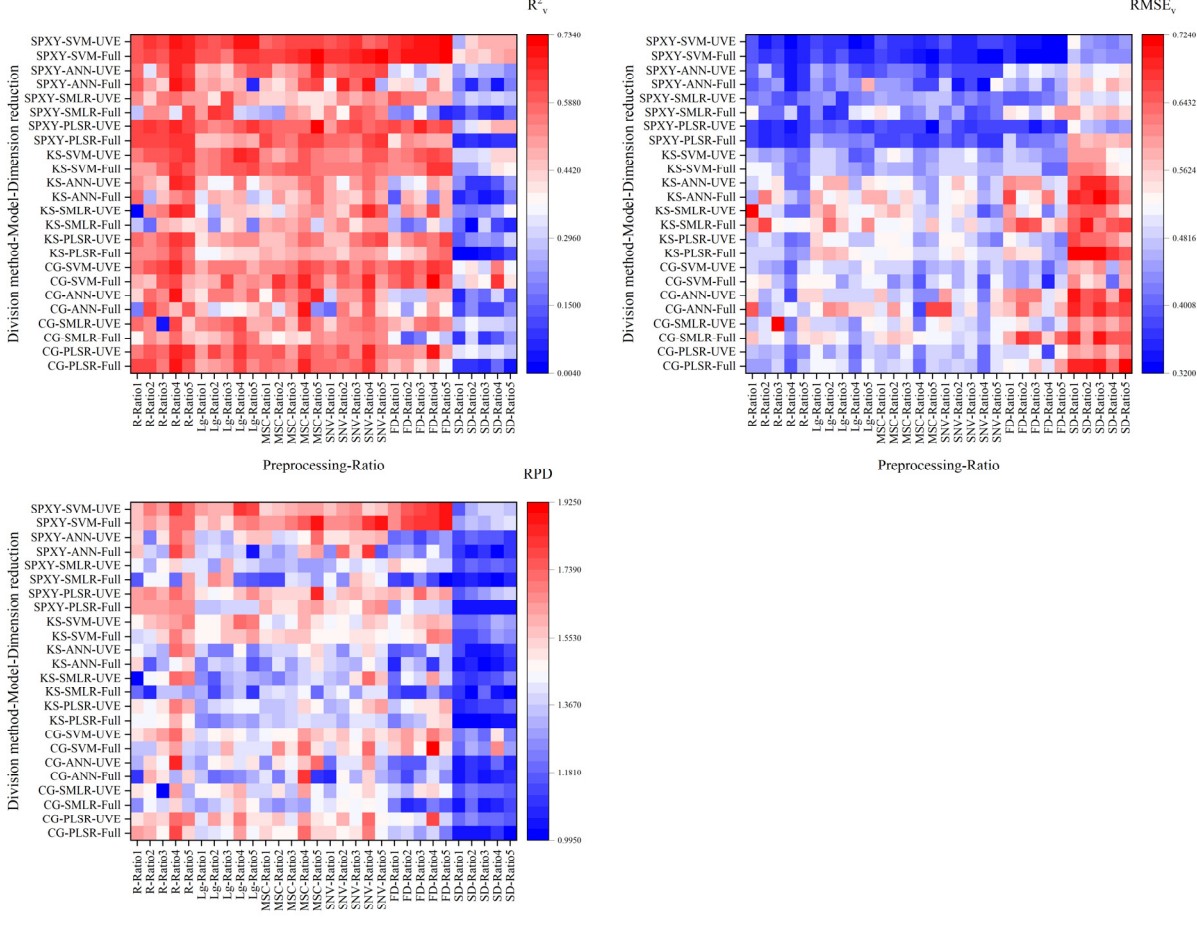

**Figure 9.** Accuracy of validation set model.

*3.6. Effect of Different Factors on Model Accuracy*

In this study, according to the correlation coefficient between each modeling effecting factor and RMSE$_v$, the effect degree of each factor on the model accuracy is analyzed. It can be seen from Table 2 that the correlation coefficient between each factor and the model accuracy reached a very significant level, indicating that these correlations had certain reliability. The order of correlation between each factor and model accuracy was the sample division method, preprocessing algorithm, sample division ratio, whether to reduce dimension, and modeling method. Among them, there was a positive correlation between the preprocessing algorithm and the model accuracy, which may be related to the low accuracy of the model based on SD spectrum. The model accuracy was negatively correlated with the sample division ratio, sample division method, modeling method, and whether to reduce dimension. This was consistent with the results that the validation accuracy was mostly the highest under Ratio4 or Ratio5, the model accuracy constructed via the SPXY method was higher than that constructed via the CG method and KS method, the SVM had higher modeling accuracy, and the model validation accuracy that was based on the UVE band was higher.

**Table 2.** Correlation between each factor and RMSE$_v$.

| | Preprocessing | Division Method | Ratio | Dimension Reduction | Model |
|---|---|---|---|---|---|
| Correlation coefficient | 0.3923 ** | −0.4629 ** | −0.1804 ** | −0.1345 ** | −0.1316 ** |

Note: ** means the significance reached 0.01 level.

## 4. Discussion

Water is an important factor affecting crop growth, and a large number of studies indicate that reasonable irrigation is an important measure to promote the normal growth of crops [41–44]. In this study, for the jointing stage, flowering stage, and later grain-filling stage, the AGDB of an irrigation treatment at these three growth stages was higher than that of a non-irrigation treatment to varying degrees, which is similar to previous research results. However, in this study, the change trend in AGDB with the growth of winter wheat in the two-year experiment was different, which may be related to the different precipitation and temperature that were experienced during the two-year experiment [45]. By analyzing the relationship between irrigation and AGDB, the water demand of winter wheat can be judged by AGDB. However, rapid acquisition of AGDB is the prerequisite for real-time judgment. At the same time, the changes in winter wheat canopy spectral reflectance were analyzed under different irrigation treatments. It was found that the spectral reflectance increased with the increase in irrigation times. This showed that irrigation can change the spectral reflectance of the winter wheat canopy by adjusting its growth. This is similar to the research results of Yang et al. [3]. This study expects to use hyperspectral technology to achieve rapid acquisition of AGDB.

Based on the analyses of previous studies, this study believes that the factors affecting the accuracy of the hyperspectral model include mainly five categories: preprocessing algorithm, sample division method, sample division ratio, whether to reduce dimensions, and modeling method. These five factors play different roles in the modeling process. It is generally believed that hyperspectral technology has the advantages of a large number of bands and a large amount of spectral information, but it also increases noise interference and spectral information redundancy [46,47]. Therefore, it is necessary to use a preprocessing algorithm to denoise the original hyperspectral data and use a dimension reduction algorithm to reduce the redundancy of spectral information. Before modeling, it is also necessary to divide all data into the calibration set and the validation set. Different sample division methods determine which samples are used as the calibration set and which samples are used as the validation set [14,30,32,33]. The sample division ratio determines the relative number of samples in the calibration set and the validation set. It is generally

believed that only the appropriate number of samples in the calibration set and the validation set can ensure the model accuracy and reduce the computational complexity at the same time [13,20]. While modeling methods are directly involved in constructing models, different modeling methods will obtain different accuracy, so they can directly affect the model accuracy [34,39,48]. When constructing the model of AGDB, the predecessors seldom considered the effect of these factors on the accuracy of the model at the same time. However, the authors of this study believed that it is necessary to consider the effect of these five factors on the accuracy of the model before constructing a hyperspectral model. Based on this, this study conducted a two-year irrigation experiment on winter wheat, collected the AGDB data and canopy hyperspectral data of winter wheat, and studied the effect of five factors on the accuracy of the hyperspectral monitoring model for the AGDB of winter wheat.

This study analyzed the role of the preprocessing algorithm in the modeling process by analyzing the characteristics of the spectral curve and its correlation with the AGDB of winter wheat. At the same time, before preprocessing, referring to the previous experience, the bands within the range of 350–399 nm, 1351–1399 nm, 1801–1950 nm, and 2451–2500 nm that were greatly affected by water and other factors shall be eliminated to preliminarily reduce the noise interference [3]. Authors of previous studies believed that five spectral preprocessing, such as Lg, MSC, SNV, FD, and SD, can reduce the noise information in the original spectrum to varying degrees [24–29]. Therefore, Lg, MSC, SNV, FD, and SD were selected to preprocess R in this study. The preprocessing results showed that—compared with R—Lg, FD, and SD preprocessing all affected the size and curve change trend in the original spectrum, while MSC and SNV preprocessing changed mainly the size of the original spectral reflectance. It can be seen that different preprocessing algorithms had different preprocessing effects on the original spectral curve, which also preliminarily indicated that different preprocessing algorithms may have different effects on reducing spectral noise interference. Correlation analysis showed that different preprocessing algorithms could improve the correlation between the spectral reflectance and the AGDB of winter wheat to varying degrees, and FD preprocessing was the best. In addition, the bands with the largest correlation between R and each preprocessing spectrum and the AGDB were 1400 nm, 1479 nm, 1083 nm, 741 nm, 797 nm, and 486 nm, respectively. These bands are important for predicting the AGDB and are close to the sensitive bands extracted by predecessors [3,7,49]. In this study, we compared the transformation spectrum with the original spectrum curve, and analyzed the role of preprocessing algorithm in improving the correlation between the original spectrum and the AGDB of winter wheat. It is believed that five spectral preprocessing algorithms, such as Lg, MSC, SNV, FD, and SD, can reduce noise interference and improve the signal-to-noise ratio to varying degrees, which was basically similar to the previous research results.

There are many sample division methods, among them, the CG method, KS method, and SPXY method are three relatively simple and widely used methods [14,50]. Therefore, these three sample division methods were selected in this study to divide the total data into the calibration set and the validation set. When predecessors studied hyperspectral models, there was no uniform standard for how many samples were used to construct models and how many samples were used to validate models. Guo et al. [51] and Hong et al. [52] used 70% of all samples to construct the model and 30% of samples to validate the model when constructing the hyperspectral monitoring model for nitrogen accumulation in winter wheat leaves and soil organic matter, respectively. However, Zhang et al. [17] chose to use 50% of all samples for modeling and the remaining 50% for validating the model when constructing the winter wheat leaf water content model. Yang et al. [20] set 5 sample division ratios of 1:1, 3:2, 2:1, 5:2, and 3:1 when constructing the hyperspectral monitoring model of soil organic carbon content. Based on this, this study also selected these five ratios when setting the sample division ratio. In this study, descriptive statistical analysis was conducted on all data sets to preliminarily analyze the role of sample division methods and ratios in the modeling process. From the results of descriptive statistical analysis,

we can see that the data distribution characteristics of different data sets were different. Both the CG method and the SPYX method divided the maximum and minimum values into calibration sets; this can ensure that the range of model validation was within the range of the calibration model, and the KS method divided only the minimum values into calibration sets. However, the results of the CG method were affected mainly by human operations [13], while the results of the KS method and the SPYX method were unique. The average value and standard deviation of all calibration sets and validation sets were close to the average value and standard deviation of the total data set, but the gap between the data set based on the SPXY method and the total data set was large, while the gap between the data set based on the CG method and the total data set was small. It can be seen from the kurtosis and skewness of each data set that most data sets were right-skewed and distributed slowly compared with the standard normal distribution. However, the validation sets that were divided by the KS method were steeper than the standard normal distribution, while the calibration set divided by the KS-Ratio1 and the validation set divided by SPXY-Ratio1, SPXY-Ratio2, and SPXY-Ratio3 all showed a certain left skewness. It can be seen that different sample division methods and sample division ratios can enrich the data characteristics of the calibration set and the validation set, which provide more possibilities for mining the internal relationships of data.

Hyperspectral technology has rich spectral information, which contains both useful and useless information. Therefore, if useful spectral bands can be screened and only useful bands are used to construct models, the prediction accuracy of models can be improved theoretically. There are many band-screening methods. Among these methods, UVE can eliminate invalid variables based on the stability of PLSR model coefficients [34,35]. Therefore, this study selected the UVE algorithm to screen useful information. It can be seen from the screening results that, based on the data sets divided by different preprocessing spectra, sample division methods, and sample division ratios, there were certain differences in the results of using UVE to screen the bands, but there were certain distributions in the visible light and short wave near-infrared range. Compared with 1852 bands in the full spectrum, 45.9503% to 99.4600% of the number of spectral bands can be eliminated by using the UVE algorithm. This paper analyzed SMLR modeling bands at the same time. The results showed that compared with the screening effect of UVE, the number of SMLR modeling bands that were based on the full-spectrum bands and UVE bands was less. Only 9.3413% and 1.5659% of the number of bands were retained at the highest value. It can be seen that using the UVE algorithm to screen bands may retain more useful bands when removing a large number of useless bands.

In this study, with the model accuracy as the final standard, the effect of the preprocessing algorithm, sample division method, sample division ratio, whether to reduce the dimension, and modeling algorithm on the hyperspectral monitoring model of winter wheat AGDB was evaluated. It can be seen from the calibration accuracy that the $R^2_c$ of most models reached more than 0.6, which preliminarily indicated that there may be a good prediction relationship between the AGDB and the hyperspectral data [39]. The preliminary analysis of different modeling factors showed that the best spectral preprocessing algorithm and sample division method was FD and KS, respectively. The best sample division ratio was 1:1, which meant that it was enough to provide a reasonable number of samples when constructing the model. Too many samples were not conducive to mining the internal relationship of data [53]. The calibration accuracy of models that were constructed based on the UVE band was generally only slightly lower than that of the models that were constructed based on the full spectrum, which showed that more simplified models and higher calibration accuracy can be obtained simultaneously after using the UVE algorithm to screen bands [34]. As far as the modeling method is concerned, SMLR and SVM were better, but it is worth noting that the $R^2_c$ and $RMSE_c$ of some models that were constructed based on SMLR were 1 and 0 kg·m$^{-2}$, respectively. The number of bands that were used in these models reached more than 110. Some researchers have believed that in multiple linear regression, the $R^2$ increased with the number of variables. Therefore, too many variables

may be the cause of high model accuracy [54], so the comprehensive performance of SVM was better than that of SMLR. Distinct from the model calibration accuracy, the best spectral preprocessing algorithm for the validation set model was R (i.e., no preprocessing), the best sample division method was SPXY, and the best sample division ratio was 5:2 or 3:1. The model validation accuracy based on the UVE band was higher than that based on the full-spectrum band. The best modeling method was SVM, which indicated that SVM itself may have better data-mining ability. At the same time, this study calculated the correlation between the accuracy of the validation set and various factors. The results showed that there was a very significant relationship between each factor and the accuracy of the model. At the same time, the positive and negative correlation also further confirmed the reliability of the above analysis on the effect of various factors on the accuracy of the model.

According to the above comparison, the theoretically optimal calibration set model should be FD-KS-Ratio1-Full-SVM, but the actual optimal model was SD-SPXY-Ratio2-Full-SVM. Theoretically, the optimal validation set model should be R-SPXY-Ratio4-UVE-SVM, but the actual optimal model was FD-CG-Ratio4-Full-SVM. It can be seen that when analyzing the effect of these modeling factors on the monitoring model of AGDB, we need to pay attention to the universality and particularity of the data at the same time. In view of its universality, a stable model with high prediction accuracy can be obtained by using the original spectrum, by dividing the calibration set and the validation set according to the ratio of 5:2 based on the SPXY method, by using UVE to screen the bands, and by using SVM to construct the model. In another study [3], we used SPA to screen bands and multiple linear regression to construct a hyperspectral monitoring model for winter wheat AGDB. The results showed that the $R^2_c$, $RMSE_c$, $R^2_v$, and $RMSE_v$ were 0.64, 0.30, 0.54, and 0.26, respectively. However, according to R-SPXY-Ratio4-UVE-SVM, we constructed the model by using the same data set, and the results showed that its $R^2_c$, $RMSE_c$, $R^2_v$, and $RMSE_v$ reached 0.8006, 0.2037, 0.8575, and 0.1567, respectively. Compared to the model in the original literature, the accuracy is higher, indicating that the model has a certain degree of reliability. This provides a strong theoretical basis for using the model to manufacture monitoring instruments for AGDB in the next step. In view of its particularity, because the calibration accuracy and validation accuracy of FD-CG-Ratio4-Full-SVM were high, its $R^2_c$, $RMSE_c$, $R^2_v$, $RMSE_v$, and RPD were 0.9487, 0.1663 kg·m$^{-2}$, 0.7335, 0.3600 kg·m$^{-2}$, and 1.9226, respectively. The validation accuracy of SD-SPXY-Ratio2-Full-SVM was low, with $R^2_v$, $RMSE_v$, and RPD values of 0.4571, 0.4328 kg·m$^{-2}$, and 1.3500, respectively. Therefore, FD-CG-Ratio4-Full-SVM was the model with the best accuracy in this study.

In this study, because the preprocessing algorithm, the sample division method, the sample division ratio, whether to reduce dimensions, and the modeling method were considered at the same time, the workload of modeling and analysis was large. Therefore, in this study, only 5 spectral preprocessing algorithms of Lg, MSC, SNV, FD, and SD; 3 sample division methods of CG, KS, and SPXY; 5 sample division ratios of 1:1, 3:2, 2:1, 5:2, and 3:1; a band-screening algorithm of UVE; and 4 modeling algorithms of PLSR, SMLR, ANN, and SVM were considered. In the subsequent research, SG smoothing, baseline correction, wavelet transform, and other preprocessing algorithms; the random method, the duplex method, the GN distance method, and other sample division methods; more scientific sample division ratio; the vegetation index, spectral characteristic parameters, PCA, and other dimension reduction algorithms; random forest, decision tree, and limit learning machine; and other modeling algorithms should be further considered to study the effect on the accuracy of the AGDB model of winter wheat. Precision agriculture and intelligent agriculture are the main development directions of agriculture at present and in the future. They all require the use of various electronic and intelligent devices to varying degrees, as these devices can help agronomists improve field management efficiency and reduce time and labor costs. For example, there are currently SPAD-502 [55], which can measure the relative chlorophyll content of crops, and Plant Canopy Analyzer [56], which can measure the leaf area index. This study aims to provide a theoretical basis for developing a device that can quickly and non-destructively measure AGDB. This study is

still at the stage of theoretical research, but through further analysis of the results of the literature [3], it can be seen that the theoretical model proposed in this study has certain reliability. In the next step, if we can further analyze the effect of five categories factors on the AGDB model of winter wheat and construct a model with higher accuracy that can be used stably, we can create an instrument that can quickly and non-destructively obtain the AGDB according to this model and provide technical support for judging the irrigation demand of winter wheat according to the AGDB.

## 5. Conclusions

This study took winter wheat as the research object; the AGDB and canopy hyperspectral reflectance of winter wheat were measured. The effects of different irrigation treatments on the AGDB of winter wheat were analyzed. In total, 5 spectral preprocessing methods of Lg, MSC, SNV, FD, and SD; 3 sample division methods of CG, KS, and SPXY; and 5 sample division ratios of 1:1, 3:2, 2:1, 5:2, and 3:1 were set up. Hyperspectral monitoring models for the AGDB of winter wheat were constructed based on the full-spectrum band and UVE band by using four modeling methods of PLSR, SMLR, ANN, and SVM. The main conclusions were as follows:

(1) Irrigation can improve the AGDB and canopy spectral reflectance of winter wheat.
(2) The preprocessing algorithm can change the original spectral curve to varying degrees and improve the correlation between the original spectrum and the AGDB of winter wheat. At the same time, 1400 nm, 1479 nm, 1083 nm, 741 nm, 797 nm, and 486 nm with high correlation with AGDB were selected.
(3) The calibration set and the validation set were divided based on different sample division methods, and sample division ratios had different data distribution characteristics.
(4) The UVE method can obviously eliminate some bands in the full-spectrum band and reduce the model complexity.
(5) In modeling algorithm, SVM had a better effect.
(6) According to the universality of the data, by using the original spectrum, combining the SPXY method to divide the samples according to the ratio of 5:2, using UVE to screen the bands, and using SVM to construct the model, we can obtain a stable and high accuracy model when using other data to construct the hyperspectral monitoring model for the AGDB of winter wheat.
(7) According to its particularity, the FD-CG-Ratio4-Full-SVM model had the highest accuracy in this study, with $R^2_c$, $RMSE_c$, $R^2_v$, $RMSE_v$, and RPD values of 0.9487, 0.1663 $kg \cdot m^{-2}$, 0.7335, 0.3600 $kg \cdot m^{-2}$, and 1.9226, respectively, which can realize hyperspectral monitoring for the AGDB of winter wheat.

**Author Contributions:** Study concept and design: M.F. and C.Y. Data analysis and drafting of the manuscript: C.Y. Experimental participants: J.X., J.B., H.S. and L.S. Critical revision of the manuscript for important intellectual content: M.F., C.W., L.X., M.Z. and X.S. Obtained funding: M.F., W.Y., C.W. and L.X. Study supervision: M.F. All authors have read and agreed to the published version of the manuscript.

**Funding:** This research was funded by Basic Research Program of Shanxi Province (20210302123411), the National Natural Science Foundation of China (31871571), the Key Technologies R & D Program of Shanxi Province (201903D211002), the Earmarked Fund for Modern Agro-industry Technology Research System (2022-07), and the Scientific and Technological Innovation Fund of Shanxi Agricultural University (2018YJ17, 2020BQ32).

**Data Availability Statement:** Not applicable.

**Acknowledgments:** We are grateful to the anonymous reviewers for their comments and recommendations.

**Conflicts of Interest:** The authors declare no conflict of interest.

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
