# Peer review of "Evaluation of Hyperspectral Monitoring Model for Aboveground Dry Biomass of Winter Wheat by Using Multiple Factors"

_agronomy, doi:10.3390/agronomy13040983_

Round 1

Reviewer 1 Report

It is hardly be considered as presented an Agronomy paper. See the attached comments. 

Author Response

Dear Reviewer:

Thank you for your valuable comments on my manuscript to make my manuscript better. The following is a detailed description of my modification of the manuscript according to your modification comments.

Comments about the manuscript titled: Evaluation of hyperspectral monitoring model for aboveground dry biomass of winter wheat using multiple factors

  • The basic idea of using new technology to achieve less laborious and time-consuming data collection about field crop could be of great significance. However, the present manuscript despite what is anticipated went in circles over doing the different statistical and mathematical presentations.

Modified. This study uses the methods that have been used many times by the predecessors, so it seems to be in circles when doing statistical and mathematical demonstrations. Therefore, in the Introduction, we explain the reasons why this study chooses the five categories factors mentioned in the text, and further explain the differences between this study and previous studies in the Discussion. Relevant contents are in lines 68-78 and 536-538.

  • The “agronomic significance presentation” is completely ignored and lost in this presentation.

Modified. In the Abstract, Introduction, and Discussion, statements about agronomic significance were added. Relevant contents are mainly in lines 34-35, 50-53, 167, and 702-707.

  • No data about actual biomass assessment presented even though it was described, mentioned.

Modified. The actual biomass changes were added to the Results, and the relevant contents were added to the Abstract, Introduction, Discussion, and Conclusion simultaneously. Relevant contents are mainly located in lines 10-11, 22, 46-53, 238-266, 506-517, 711, and 717.

  • Most of the literature referred to is of different kind and approaches. For example the remote sensing of that literature was via satellite. However it is used and referred to as if it is not different than the present work.

Modified. The purpose of this study is to realize the rapid measurement of winter wheat aboveground dry biomass using hyperspectral technology. Therefore, the effect of the factors of different preprocessing algorithms, sample division methods, sample division ratios, dimensionality reduction algorithms, and modeling algorithms on the aboveground dry biomass model of winter wheat was analyzed in this study. The literature we refer to in this paper only focused on the effect of one, two, or three of the above five factors on the model. However, according to the process of constructing models using hyperspectral data, I think the above five factors are all very important for obtaining models with high accuracy. This is also the biggest difference between this study and previous studies, even though the same methods are used to different degrees. The content added in lines 68-78 and 536-538, combined with the content in lines 79-84 of the original text, further expounds the necessity of carrying out this study and the difference between this study and previous studies.

  • Table 1, not clearly presented. Not clearly comprehended, and not discussed.

Modified. The analysis in Table 1 has been further modified in the Results and Discussion. The relevant contents are in lines 316-338 and 595-602.

  • The rest figures sound like just cosmetic presentation. It has no “Agronomic” merit.

Modified. Hyperspectral technology has the advantage of fast and nondestructive acquisition of target features. The figures in this paper can reflect the role of some modeling factors in the construction of aboveground dry biomass model. If a model with high accuracy and stability can be constructed, it can serve for rapid acquisition of aboveground dry biomass. This can provide technical support for using aboveground dry biomass to determine crop water demand. Add relevant content in lines 702-707 to further elaborate the relationship between these figures and agronomy.

  • Discussion, very wordy and once more presented no agronomy merit.

Modified. The unimportant part of the Discussion was deleted, and the content of the agronomy merit was added in lines 702-707.

  • It would have been mush more direct to the point and of more merit to agronomist to summarize the whole discussion around the points presented in the conclusion. However, per example point 1 is left so vague and not specific.

Modified. According to the content in the Conclusion, we have simplified the content in the Discussion and strengthened the narration of agronomic significance.

  • All the way through points 1-6 of the conclusion a significant and central question is raised; what in hand an agronomist have on hand as far as the practical application of this technique ??

Modified. At present, this study is still in the theoretical research stage and serves to construct a more accurate aboveground dry biomass model. If a high-accuracy model can be found, an instrument that can rapidly measure the aboveground dry biomass of winter wheat can be developed based on spectral technology. At that time, agronomists can use this instrument to quickly obtain the aboveground dry biomass of winter wheat, providing reference for field irrigation, fertilization, and other measures. Relevant contents are in lines 702-707.

  • The over all impression about this work hinges about the possibility that it is done by “non agronomists” some technology appropriators.

Modified. The research basis of this study is an irrigation experiment of winter wheat. We added the content about the effect of irrigation on the winter wheat aboveground dry biomass change to strengthen the connection between this study and agronomy.

  • The final conclusion about its feasibility to be published as an “agronomy” piece of work cannot be positive. On the contrary it may be positive to others. So for “Agronomy Journal” it has to be look upon twice and weigh that according to relatedness. Hence, It is left up to the Journal’s editorial board to accept or reject its acceptance. However if the board accept such work publication the manuscript has to take measures in making the above point raised taking care of and re-orient the paper toward agronomy work.

Modified. Thank you very much for giving me a chance to revise my paper, so that it may be accepted by "Agronomy". We added the analysis of the aboveground dry biomass change rule to strengthen the correlation with agronomy, and added the elaboration of the agronomic significance.

Reviewer 2 Report

General comments

I think this paper tries to do too much. The results are extremely hard to understand, given the dense language, the proliferation of acronyms and the myriad of models, approaches and methods. This is really four papers crammed into one. This does not mean that the substance is not publishable – it just needs a lot of reworking and simplifying.

One thing that was very confusing was the lack of a time domain. Are you averaging the total biomass through time? It says on line 162 that each plot was ‘measured’ 10 times – where is this data? Does this mean that there were ten spectral curves that were averaged to create the datatset in Figure 1? No where is that stated.  It is extremely unclear what is being correlated to what and the number of observations, etc. Much greater clarity is needed to help the reader understand what was done.

Specific comments:

Abstract – The spectral bands from 45.9503% to 99.4600% - what does this mean? Are these band wavelengths? Please rephrase or be more general in the abstract. Assume the reader knows nothing about your instrument or sensor type for the abstract text.

Lines 98-101 – this is exactly what is stated in lines 63-65 – rephrase or simply say ‘the five categories’ since you defined this above.

It is very hard to read the introduction. Please add a few paragraph breaks to the section from 59 to 101 and from 102 to 140. For the first section I suggest line 71 and 88. For the second section I suggest at line 114 and line 129.

Lines 129-135 The purpose is to… Capitalization after a semi-colon is not needed and is confusing. Why don’t you simply put periods instead of semi-colons, so that it makes sense. Also your bullet three is not parallel with the sentence construction of items 1 and 2.

Figure 1 caption – please define all your acronyms. You have so many, help the reader remember them by defining them in the caption. Also, which figure has the ‘original spectrum’?

Text in the results section should be in present tense, not past tense.

Lines 223-224 state that the figure shows the correlation between R and preprocessing spectra and above-  ground dry biomass of winter wheat. - I really don’t understand this. I think what the authors have done is to take the spectral measurement of each wheat experiment, which was then destructively sampled, dried and weighed. If this was the case, what does Figure 2 show? I cannot see how each spectral band is correlated with just one number per location, which is what the ‘above ground biomass of winter wheat’ would be.  I think these correlation result should be reported in a table with significance and P values, etc.

Author Response

Dear Reviewer:

Thank you for your valuable comments on my manuscript to make my manuscript better. The following is a detailed description of my modification of the manuscript according to your modification comments.

General comments

I think this paper tries to do too much. The results are extremely hard to understand, given the dense language, the proliferation of acronyms and the myriad of models, approaches and methods. This is really four papers crammed into one. This does not mean that the substance is not publishable – it just needs a lot of reworking and simplifying.

Modified. We have reworked and simplified the manuscript according to the comments of reviewers.

One thing that was very confusing was the lack of a time domain. Are you averaging the total biomass through time? It says on line 162 that each plot was ‘measured’ 10 times – where is this data? Does this mean that there were ten spectral curves that were averaged to create the datatset in Figure 1? No where is that stated.  It is extremely unclear what is being correlated to what and the number of observations, etc. Much greater clarity is needed to help the reader understand what was done.

Modified. Our expression in the text is not clear enough, which leads to ambiguity. What we want to express is that for a certain plot, we collected 10 spectral curves (because we need to avoid the unreliability caused by only measuring once), and calculated the average value of these 10 curves, taking the average value as the canopy spectrum of winter wheat in the plot. It is measured in this way for each plot in each growth stage. The relevant content has been modified, at lines 189-192.

The spectral curve data in Figure 2 (original Figure 1) comes from the sample with aboveground dry biomass of 1.3405 kg·m-2. Since the aboveground dry biomass of this sample is closest to the average of the aboveground dry biomass of all samples in this study, so the spectrum of this sample is taken as an example for analysis. Relevant content has been added in lines 269-271 to make the manuscript easier to understand.

Specific comments:

Abstract – The spectral bands from 45.9503% to 99.4600% - what does this mean? Are these band wavelengths? Please rephrase or be more general in the abstract. Assume the reader knows nothing about your instrument or sensor type for the abstract text.

Modified. 45.9503% - 99.4600% refers to the number of bands, that is, 45.9503% - 99.4600% of the number of full spectrum bands. The relevant content in the full manuscript has been modified, and the relevant content is located in lines 26, 624, and 629.

Lines 98-101 – this is exactly what is stated in lines 63-65 – rephrase or simply say ‘the five categories’ since you defined this above.

Modified. The original text is indeed cumbersome, and the relevant content after modification is located at line 121.

It is very hard to read the introduction. Please add a few paragraph breaks to the section from 59 to 101 and from 102 to 140. For the first section I suggest line 71 and 88. For the second section I suggest at line 114 and line 129.

Modified. According to your suggestion, paragraph breaks have been added to the relevant position.

Lines 129-135 The purpose is to… Capitalization after a semi-colon is not needed and is confusing. Why don’t you simply put periods instead of semi-colons, so that it makes sense. Also your bullet three is not parallel with the sentence construction of items 1 and 2.

Modified. We use periods instead of semi-colons. The research purpose is rewritten according to the unified sentence structure. Relevant contents are in lines 155-160.

Figure 1 caption – please define all your acronyms. You have so many, help the reader remember them by defining them in the caption. Also, which figure has the ‘original spectrum’?

Modified. A note is added after the caption of Figure 2 (original Figure 1) to explain the abbreviation. The relevant contents are in lines 288-291. Where R represents the original spectrum.

Text in the results section should be in present tense, not past tense.

Modified. We asked our colleagues to revise the writing tense of the full text and change some of the text in results to the present tense.

Lines 223-224 state that the figure shows the correlation between R and preprocessing spectra and above-  ground dry biomass of winter wheat. - I really don’t understand this. I think what the authors have done is to take the spectral measurement of each wheat experiment, which was then destructively sampled, dried and weighed. If this was the case, what does Figure 2 show? I cannot see how each spectral band is correlated with just one number per location, which is what the ‘above ground biomass of winter wheat’ would be.  I think these correlation result should be reported in a table with significance and P values, etc.

Modified. In this study, each sample has one aboveground dry biomass value and corresponding hyperspectral reflectance curve, that is, 240 samples have 240 aboveground dry biomass values and corresponding hyperspectral reflectance curves. Because of the good correlation between them, we hope to preliminarily explain why hyperspectral technology can be used to monitor the aboveground dry biomass of winter wheat through correlation analysis. At the same time, for different preprocessing spectra, because the correlation between different spectra and aboveground dry biomass is different, correlation analysis can preliminarily evaluate the role of preprocessing algorithms in constructing hyperspectral models.

In addition, since the spectral curve contains more than 2000 bands, the correlation between each band and aboveground dry biomass may be different, so we believe that the results cannot be effectively displayed using tables. However, according to your opinion, we have added a correlation critical value line (p<0.05) in Figure 3 (original Figure 2). From the figure, we can also see that there are differences in the position of bands where different spectra and aboveground dry biomass reach significant correlation, which once again shows the different roles of preprocessing algorithms in constructing spectral models.

Reviewer 3 Report

The manuscript entitled “Evaluation of hyperspectral monitoring model for aboveground dry biomass of winter wheat using multiple factors” presents interesting study on remote sensing application in crop production. The study is quite interesting and the manuscript is quite well prepared.

Below are detailed comments on the manuscript.

1. Please provide more details about data acquisition. For example there is lack information what is the view of Field-Spec 3.0 spectrometer. Please provide information about diameter size which is evaluated. It is probably quite small (several cm?). How to avoid effect of evaluation of small area? The area for which the reflectance was measured should be representative for total crop canopy.

How many bands/variables were registered? What was the range of individual band, how many nm?

2. The figures should be self-explanatory. What is presented as the error bars in Fig. 1? It is standard deviation or other parameter? What method of multiple comparison of the means was used?

3. Line 242: it should be “significance level” not “significant level”.

4. It would be good if you present the raw data used for the analyses, i.e. reflectance profiles for each treatment. It will allow to evaluate the differences visually.

5. Are the results in Fig. 2 and 3 for one treatment? For what growth stage of the crop? It is not clear what data are presented.

6. In such study is important which bands (wavelengths) of the reflectance have strongest relationship with AGDB. Could you provide the results which inform which bands are most important for evaluation of AGDB?

7. Please adjust formatting of the references to the requirements of the journal, e.g. years should be in bold, and names of the journals should be abbreviated.

Author Response

Dear Reviewer:

Thank you for your valuable comments on my manuscript to make my manuscript better. The following is a detailed description of my modification of the manuscript according to your modification comments.

The manuscript entitled “Evaluation of hyperspectral monitoring model for aboveground dry biomass of winter wheat using multiple factors” presents interesting study on remote sensing application in crop production. The study is quite interesting and the manuscript is quite well prepared.

Below are detailed comments on the manuscript.

  1. Please provide more details about data acquisition. For example there is lack information what is the view of Field-Spec 3.0 spectrometer. Please provide information about diameter size which is evaluated. It is probably quite small (several cm?). How to avoid effect of evaluation of small area? The area for which the reflectance was measured should be representative for total crop canopy.

How many bands/variables were registered? What was the range of individual band, how many nm?

Modified. In this study, when measuring the canopy spectrum using the Field Spec3.0 spectrometer, the probe with a field angle of 25 ° was selected. That is, when the probe is 1m away from the canopy, the diameter of the collection area is about 44.34cm, and the spectrum of three rows of winter wheat can be collected (the row spacing of wheat planting in this study is 20 cm). At the same time, for a certain plot, select a position with uniform growth to collect the spectrum, so that the measured spectrum can represent the spectrum of the whole plot.

In addition, the spectral sampling range, spectral sampling interval, and spectral resolution of the Field spec 3.0 spectrometer used in this study were added.

The relevant modifications are in lines 170-176.

  1. The figures should be self-explanatory. What is presented as the error bars in Fig. 1? It is standard deviation or other parameter? What method of multiple comparison of the means was used?

Modified. The error bar in Figure 1 represents the standard deviation. Tukey method is used for multiple comparison. Relevant content is added in line 224.

  1. Line 242: it should be “significance level” not “significant level”.

Modified. Modified at line 225.

  1. It would be good if you present the raw data used for the analyses, i.e. reflectance profiles for each treatment. It will allow to evaluate the differences visually.

Modified. Taking the data of flowering stage in 2021 as an example, the contents of spectral reflectance curves of different treatments are added in section 3.1. Relevant contents are in lines 251-267.

  1. Are the results in Fig. 2 and 3 for one treatment? For what growth stage of the crop? It is not clear what data are presented.

Modified. The spectral curve in Figure 3 (original Figure 2) takes T3 treatment at flowering stage in 2021 as an example, because the AGDB of this sample is 1.3405 kg·m-2, which is closest to the average value of AGDB of all samples in this study. It has been modified in the text, and the relevant contents are in lines 270-272. Figure 4 (original Figure 3) is obtained by correlation analysis of AGDB and corresponding spectral data of all samples. The expression in the text has been modified, and the relevant contents are in lines 294-295.

  1. In such study is important which bands (wavelengths) of the reflectance have strongest relationship with AGDB. Could you provide the results which inform which bands are most important for evaluation of AGDB?

Modified. When analyzing the role of the preprocessing algorithm in the modeling process, we have found that the band positions when the correlation between R, Lg, MSC, SNV, FD, and SD spectra and AGDB reached the highest were 1400 nm, 1479 nm, 1083 nm, 741 nm, 797 nm, and 486 nm, respectively. According to previous studies, these bands play an important role in monitoring winter wheat AGDB. We have further added relevant results to the Abstract and Conclusion. Relevant contents are in lines 23-24 and 676-677.

  1. Please adjust formatting of the references to the requirements of the journal, e.g. years should be in bold, and names of the journals should be abbreviated.

Modified. The format of references was checked and revised to the format required by the journal.

Round 2

Reviewer 1 Report

I read your response to the raised comments. I see you made changes according to most of the point raised on the track editing copy. However, some were not accepted or just went around such as the comment about the figures. It is acknowledged that the present work and so far is a preliminary hypothetical then it needs further application to be more related to agronomy and the agronomists.

Author Response

Dear Reviewer:

Thank you for your valuable comments on my manuscript to make my manuscript better. We have made further modifications to the manuscript based on your comments. The following is a detailed description of my modification of the manuscript according to your modification comments.

I read your response to the raised comments. I see you made changes according to most of the point raised on the track editing copy. However, some were not accepted or just went around such as the comment about the figures. It is acknowledged that the present work and so far is a preliminary hypothetical then it needs further application to be more related to agronomy and the agronomists.

Further modifications have been made. In the Discussion section, we further constructed and validated the model(R-SPXY-Ratio4-UVE-SVM) using data from a literature published in another journal. The results showed that compared to the accuracy of the original model, the reconstructed model had a higher accuracy. This indicated that the model proposed in this study has certain reliability and universality. Devices play an important role in both current and future agriculture. These devices can help agronomists judge the growth status of crops and provide management efficiency. This research is currently in the theoretical research stage, but obtaining a model with high reliability and universality is an important prerequisite for developing real-time measurement AGDB devices. In addition, in fact, many of the figures in this paper are not directly related to agronomy, but the content of these figures is intended to construct AGDB models with high reliability and universality, so they have an indirect role in promoting the development of agronomy. Relevant content has been added in lines 721-729 and 752-761.

Reviewer 2 Report

The paper is improved. I think that a simpler presentation would increase its utility and citation rate, but it is acceptable as it is. I recommend that the authors consider a more incremental publishing approach, which is more accessible to an interdisciplinary audience. 

The revisions have added a number of typos and editorial issues that the journal will point out during type setting. I suggest two edits below. 

This sentence in the abstract still doesn't say anything - 'The number of full spectral bands from 45.9503% to 99.4600% can be eliminated by UVE method.'

Please revise to simply remove 'from 45.9503% to 99.4600%'.  without more information, these percentages have no meaning. You are saying that the approach reduces the bands needed, right? 

Figure 1 - please include more information in the caption. What is T1? T2? define your acronyms.   Also in the note, please state that a denotes significance, whereas b denotes... not sure what, but please clarify.

Author Response

Dear Reviewer:

Thank you for your valuable comments on my manuscript to make my manuscript better. We have made further modifications to the manuscript based on your comments. The following is a detailed description of my modification of the manuscript according to your modification comments.

The paper is improved. I think that a simpler presentation would increase its utility and citation rate, but it is acceptable as it is. I recommend that the authors consider a more incremental publishing approach, which is more accessible to an interdisciplinary audience. 

The revisions have added a number of typos and editorial issues that the journal will point out during type setting. I suggest two edits below.

This sentence in the abstract still doesn't say anything - 'The number of full spectral bands from 45.9503% to 99.4600% can be eliminated by UVE method.'

Please revise to simply remove 'from 45.9503% to 99.4600%'.  without more information, these percentages have no meaning. You are saying that the approach reduces the bands needed, right? 

Modified. ‘From 45.9503% to 99.4600%' has been deleted from the Abstract and Conclusions. The relevant content is located in lines 27-28 and 784-785.

Figure 1 - please include more information in the caption. What is T1? T2? define your acronyms.   Also in the note, please state that a denotes significance, whereas b denotes... not sure what, but please clarify.

Modified. Explanations for T1, T2, T3, T4, and T5 have been added to the note. At the same time, the lowercase letters that indicate the significance of the difference in the figure were explained in detail. The relevant content is located in lines 261-267.

Reviewer 3 Report

The manuscript was improved according all my suggestions and current version of the manuscript can be published.

Author Response

Thank you for your approval of my paper. Thank you again for your work.